# Unsupervised Dynamic Routing Via Competition Over Network Loss

## Abstract

This paper proposes a novel neural network architecture that can simultaneously do normal network optimizing while attaining the ability of unsupervised learning. Almost all existing unsupervised learning algorithms are based on doing calculations on the input space or feature space, this paper proposes a new possibility to discover a structure in the functional space without supervision. Using the self-organizing map over the competition of the loss of individual neural column, we route the input to the most appropriate modules dynamically, by doing this we separate the input functional space into different sub spaces which are represented by each individual neural column. At the end of the paper, we propose several possible architectures based on the philosophy of this paper that could build a neural network system block by block.

## 1 Introduction

The unsupervised learning ability of the natural structured biological neural network is always fascinating scientists. The structure of some primitive neural networks is generated from evolution [1], while other superior networks have a base neural substrate that could adapt to the input neural signal and generate a suitable structure automatically[2][3][4]. Even computer simulation of this process gives the same result. By simulating the evolution process by genetic algorithm, a scientist could evolve the same network structure for the insect's path integration[5]. On the same subject, one could also use a general purpose recurrent neural network (RNN) to optimize the path integration problem and still get the same network structure by interpreting the network weights[6][7]. This method still works for more complicated problems, some scientists could replicate the neuron activation pattern for the grid cell in the hippocampus of the rat, which is used for rat's navigation and path integration[8][9][10]. Although different in optimizing algorithm and network structure, RNN and hippocampus could get the same activation pattern eventually on the subject of navigation which means that to develop an artificial network structure to mimic the biological plausible structure is not impossible.

Information has its own structure and the structure is important. Convolutional neural network (CNN) works best in the domain of image information[11] while natural language sequence information works best under the structure of recurrent neural network (RNN) or deep self attention network (Transformer)[12]. Image information has a structural match with the structure of CNN, so even without training the structure of CNN could give us a relatively high accuracy in classification task which is better than random guessing[13]. So the hierarchical structure of image information and the hierarchical network structure of CNN is having a resonance. On the other hand, natural language sequence doesn't have such a strict information structure, the "pixels" of the sequence has a

Submitted to 35th Conference on Neural Information Processing Systems (NeurIPS 2021). Do not distribute.

correlation of each other some random distance away, so the self attention algorithm works best under this scenario[12]. Can any neural network learn such information structure automatically that it can adapt to any kind of information universally? The biological neural network of vertebrate animals seems achieved this goal already by evolved a columnar structure[3], so we continue to adopt this philosophy on the design of neural network architecture.

**Contribution**   We proposed a novel network architecture called functional self organizing map (fSOM) for making connection between network layers and also a method to discover the connection target automatically by dynamic routing among the neural columns between layers by mutual competition between network columns.

## 2   Related Work

**Capsule networks**   For the area of columnar networks that can do dynamic routing, some method has been proposed[14][16]. Hinton first introduced the concept of capsules which is a multi-layer columnar network which are trained to generated images with certain transformations. However in their method, if the feature size is large then the mutual connection matrix will get large quickly which will make the optimization slow. Actually if considering employing regularization to enhance the generalization of the network, most of the number in the feature vector is zero with only a small portion of the vector non zero after activation function[15]. As seen in Fig1 in appendix we can partition the feature vector into columns and only transfer this non-zero column to the next layer. And also, their method is based on a local routing rule of *mutual agreement* which although to some extend have a biological plausibility, doesn't take into account the mutual competition of the columns involved so the network lacks of a global interpretability because the columns it generated don't have a topological relationship.

## 3   Functional Self Organizing Map

### 3.1   Introduction to self organizing map

Here we give an introduction to the unsupervised learning algorithm of the self organizing map (SOM) which is selected as the mutual competition algorithm of our method[18]. Each data from our dataset is a vector $\mathbf{X}^m \in \mathbf{R}^m$ of dimension $m$. The SOM is composed of a group of vectors $\mathbf{W}_i^m$(also called the weight of the SOM) which have the same dimension $m$ that is distributed on a two dimensional map. The index $i$ of $\mathbf{W}_i^m$ is the position of the weight vectors as seen in Fig 1. The input vector is then compared with each of the vectors on the map, the one with the minimum distance $d$ from the input vector will be selected for update, which is called the best matching unit (BMU):

$$d_i = \frac{1}{m}\sqrt{\sum_m (\mathbf{X}^m - \mathbf{W}_i^m)^2} \tag{1}$$

$$BMU^m = \mathbf{W}_{\arg\min_i(d_i)}^m \tag{2}$$

where $d_i$ is called distance function, $BMU^m$ is the best matching unit vector. To update the weight, we need to make the vector of the BMU closer to the input vector according to hebbian learning rule[19]. We also need to update the vectors that are close to BMU, otherwise, the same BMU will always be selected no matter what is input. To do this, we need to define a neighborhood function of the BMU and assign different rates when updating the weight.

$$h_i = \alpha \exp(-\frac{\|r_i - r_{BMU}\|^2}{2\sigma^2}) \tag{3}$$

Where $h_i$ is called the neighborhood function, $r_i \in \mathbf{R}^2$ and $r_{BMU} \in \mathbf{R}^2$ are the position coordinates on the SOM, $\sigma$ is the radius of the neighborhood function. The weights of the SOM will be updated according to the following rule.

$$\mathbf{W}_i^{m\prime} = \mathbf{W}_i^m + h_i(\mathbf{X}^m - \mathbf{W}_i^m)$$

This means we can scatter the input representation space on a map $\mathbf{X}^m \Rightarrow \mathbf{W}_i^m$. We'll show how to regress some output space to some input space $G(\mathbf{X}^m) = \mathbf{Y}^n$ by scattering this functional space topologically on a map $(\mathbf{X}^m, \mathbf{Y}^n) \Rightarrow \mathbf{G}^{mn}$ as we did on SOM.

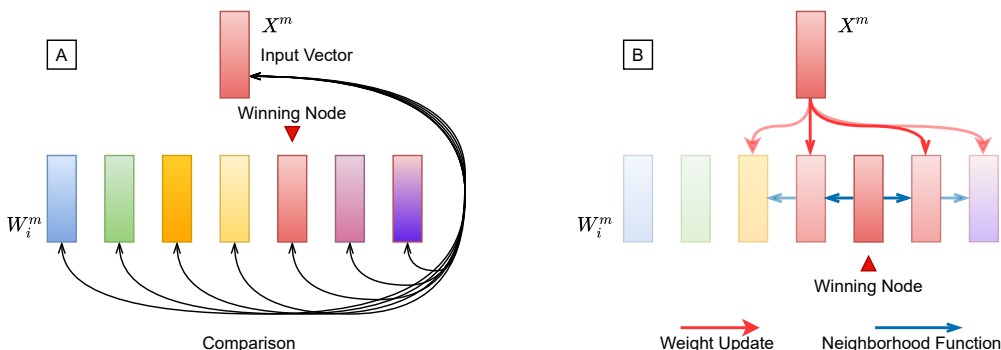

Figure 1: A) Input vector $\mathbf{X}^m$ is compared with every node vector on the SOM $\mathbf{W_i^m}$, the node with minimum distance $d$ is selected for the best matching unit (BMU). B) Each node on the SOM is updated following the input vector. The node of the BMU and other node that is close to the BMU have a larger learning rate while other nodes have lower learning rate according to the neighborhood function.

## 3.2 Functional Self Organizing Map

The main idea of the SOM is competition. By competing over every other node, we get a winning node and update the node on that position. Adopting the same idea, instead of using vectors, we use neural network columns to compose the SOM 2. Each column is a two layer encoder-decoder unit. For an input vector $\mathbf{X}^m$, each of the columns will give an output vector as follows:

$$\mathbf{Y}_i^n = \sum_{ml} \mathbf{X}^m \mathbf{E}_i^{ml} \mathbf{D}_i^{ln}$$

Where $\mathbf{X}^m$ is the input vector, $\mathbf{E}_i^{ml}$ is the encoder matrix of column $i$, $\mathbf{D}_i^{ln}$ is the decoder matrix and $\mathbf{Y}_i^n$ is the output vector of dimension $n$. Note no activation function and weight regularization are needed as explained in Fig1 of appendix. Now we can define the loss function of each column to be:

$$\mathbf{L}_i = \frac{1}{n} \sum_n |\mathbf{T}^n - \mathbf{Y}_i^n| \tag{4}$$

$$BMU\ index = \arg\min_i(L_i)$$

where $\mathbf{T}^n$ is the target vector. Then the node with minimum loss is selected as the BMU and has the priority to fully back propagate the error. Other error back propagation is penalized by the value of the neighborhood function by limiting the loss value associated with that node, so the overall gradient needs to propagate back is:

$$\nabla_\theta \sum_i h_i \mathbf{L}_i$$

$$h_i = \exp(-\frac{\|r_i - r_{BMU}\|^2}{2\sigma^2})$$

where $\theta$ is the parameters of all the columns. Note that not all columns can propagate gradient back, only those within some certain radius of the BMU have the opportunity as seen in Fig 2. At this stage, there's no mutual connection between columns, so different columns will be optimized to different directions although the loss function could be the same. To make columns more coherent with their surrounding, we need to add a coherence update to the parameters of the columns. Now define the

optimized parameters of the network to be $\hat{\mathbf{E}}^{ml}$ and $\hat{\mathbf{D}}^{ln}$, we can make coherence update like the following:

$$\mathbf{E}_i^{ml\prime} = \mathbf{E}_i^{ml} + \alpha h_i(\hat{\mathbf{E}}^{ml} - \mathbf{E}_i^{ml})$$

$$\mathbf{D}_i^{ln\prime} = \mathbf{D}_i^{ln} + \alpha h_i(\hat{\mathbf{D}}^{ln} - \mathbf{D}_i^{ln})$$

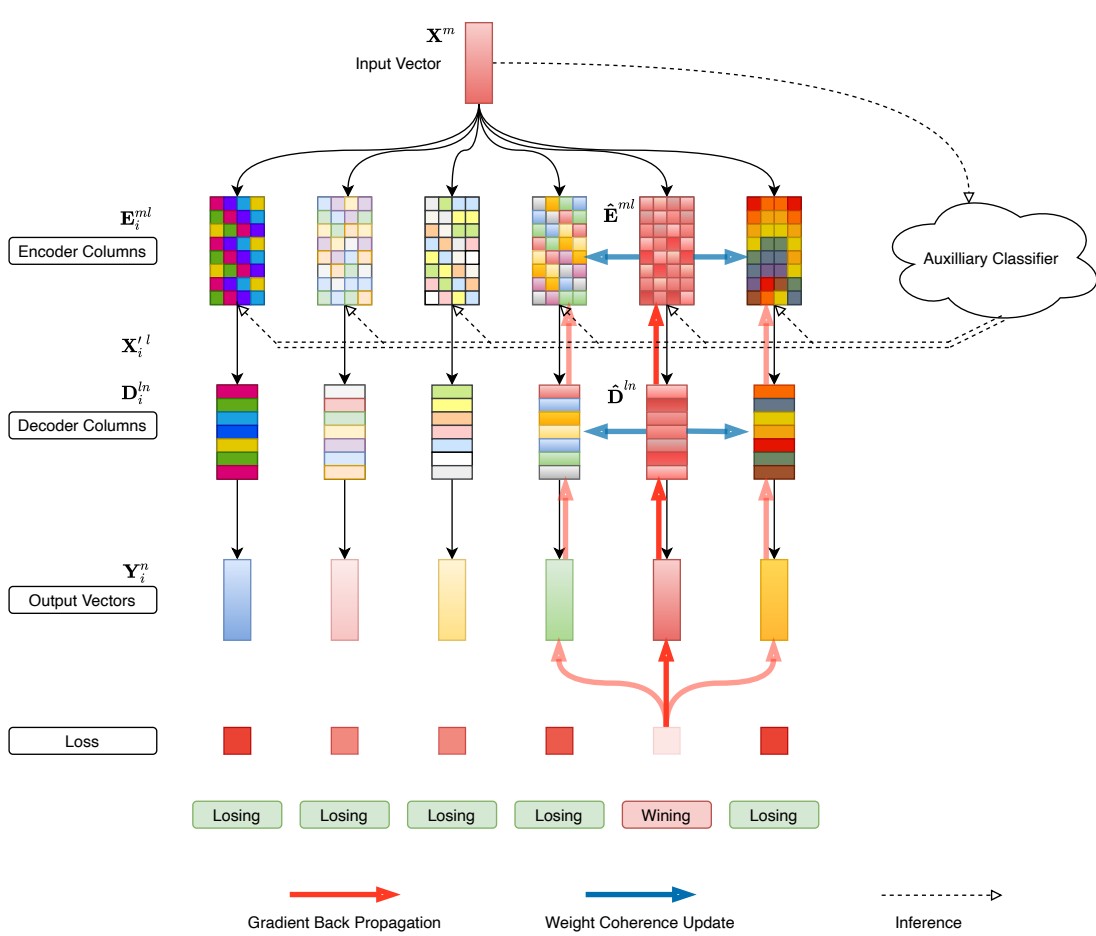

Figure 2: Functional Self Organizing Map, please see section 3.2 for detail

where $\alpha$ is the coherence update rate. This means, we are going to have a topologically functional space composed of each column, and also we dynamically routed the input to output by doing a winner-take-all competition. Note that during optimization, not all the columns are optimized by back propagation, only those surrounding BMU are. So to make an inference on the network we need to record the chosen BMU location and train an auxiliary classifier as seen in Fig 2, then by using this classifier we can select the correct column to output. By doing this, the scope of the calculation is only limited to the classifier and the selected column during inference.

## 3.3 Multi-layered Functional Self Organizing Map

For multi-layered networks, we need a dynamic routing algorithm between layers. In other works[14] columns between different layers are fully connected which poses a problem such that we need

every loss of the columns to find the BMU and do the optimization. A multi-layered network has a combinatorial number of losses to calculate and the number of losses will grow exponentially large. Here we adopt a simplification strategy by expanding the fully connected layers to a hierarchical cascade network. Fig 3 shows a two layered functional SOM network. The output is calculated by columns in a different layer as following:

$$\mathbf{Y}_i^n = \sum_{mlk} \mathbf{X}^m \mathbf{E}_{\hat{j}}^{ml} \mathbf{F}_i^{lk} \mathbf{D}_i^{kn}$$

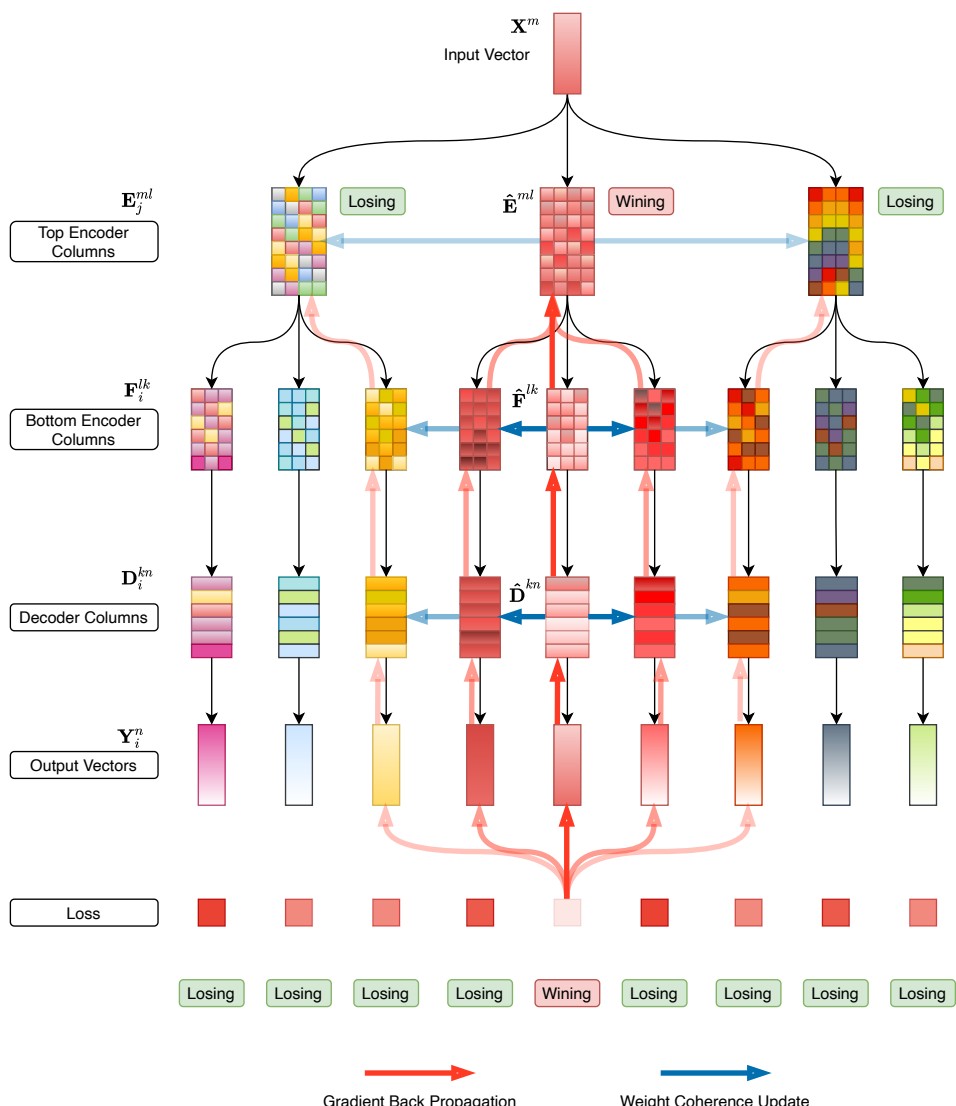

Figure 3: Multi-layered Functional Self Organizing Map, please see section 3.3 for detail

Where $\hat{j}$ is the selected index of the first layer. The calculations to find the distance function and BMU is the same as in Equation 1 2. However for there are two layers, we need two neighborhood functions:

$$h_i = \exp(-\frac{\|r_i - r_i^{BMU}\|^2}{2\sigma_1^2}) \quad h_j = \exp(-\frac{\|r_j - r_j^{BMU}\|^2}{2\sigma_2^2})$$

Where $\sigma_1\sigma_2$ are the neighborhood radius of each layer, $r_i^{BMU}, r_j^{BMU}$ are the position coordinates of the BMU. To make gradient back propagation work as in the single layer case, we need to apply the neighborhood function to the loss. Different layers have different map sizes, so the neighborhood function of each layer has a different granular scale. In Fig 3, each top layer node connects to three bottom layer nodes so the neighborhood function needs to triple the size to match the bottom layer with the scale-up function $upsize^3()$, the resulting neighborhood function is the sum of the two:

$$h_i^* = \exp(-\frac{\|r_i - r_i^{BMU}\|^2}{2\sigma_1^2}) + \beta upsize^3(\exp(-\frac{\|r_j - r_j^{BMU}\|^2}{2\sigma_2^2})) \tag{5}$$

where $\beta$ controls the relative ratio of the top layer neighborhood function. And the gradient back propagation is the same:

$$\nabla_\theta \sum_i h_i^* \mathbf{L}_i$$

As in the single layer case, we still need a coherence update to all the column parameters:

$$\mathbf{E}_j^{ml\prime} = \mathbf{E}_j^{ml} + \alpha h_j(\hat{\mathbf{E}}^{ml} - \mathbf{E}_j^{ml})$$
$$\mathbf{F}_i^{lk\prime} = \mathbf{F}_i^{lk} + \alpha h_i(\hat{\mathbf{F}}^{lk} - \mathbf{F}_i^{lk})$$
$$\mathbf{D}_i^{kn\prime} = \mathbf{D}_i^{kn} + \alpha h_i(\hat{\mathbf{D}}^{kn} - \mathbf{D}_i^{kn})$$

After the optimization, we are going to have a coarse functional map for the top layer and a finer map belongs to each top node for the bottom layer. We'll show the generated map in the next section.

# 4 Experiments

All the experiments are conducted on one GPU machine of 2080Ti. The code will be published in Github online repository[1].

## 4.1 One Layer Functional Self Organizing Map

We first test our method on the one-layer case explained in Section 3.2. The dataset we used is the handwritten digit dataset *mnist* [20]. The training target $\mathbf{T}^n$ in Equation 4 is simply the same as input vector $\mathbf{X}^m$ which makes the optimization into an auto-encoder regression. All hyperparameters are summarized in the following table.

Table 1: Single layer network hyper parameters

| Phase | Map size | Radius | Adam learning rate | Coherence update rate |
|-------|----------|--------|--------------------|------------------------|
| Start | 16 | 2.0 | $1e^{-4}$ | 1.0 |
| End | 16 | 0.5 | $1e^{-4}$ | 0.1 |

We terminated the optimization after the decoder weight is stabilized and generated the figure of decoder weights for visualization. Please see Figure 4 for detail.

## 4.2 Two Layer Functional Self Organizing Map

Secondly, we test our multi-layer method on the same dataset of *mnist*. The training target is again an auto-encoder regression.

We generated the figure of the top encoder and decoder weights, the bottom encoder is not visually interpretable, so we just show one sample of it. We also generated the overall neighborhood function defined in Equation 5. We then test the BMU indices of the top layer against the true label to check

---

[1]https://github.com/threesond/fSOM

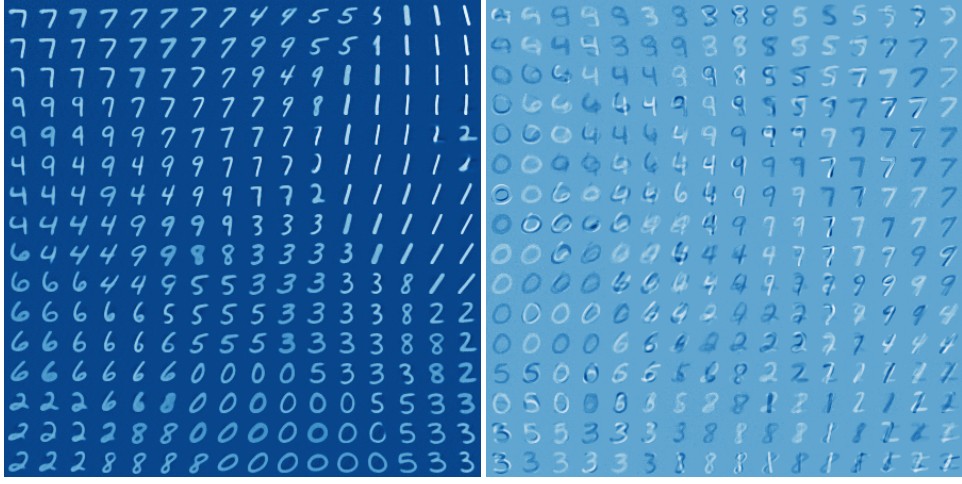

(a) Decoder map with coherence update      (b) Decoder map *without* coherence update

Figure 4: Single layer experiments result. Selected encoder and decoder channel is tiled on a $16 \times 16$ grid.

Table 2: Single layer network hyper parameters

| Phase | Top map size | Bottom map size | Top radius | Bottom radius | Adam learning rate | Coherence update rate | Relative ratio |
|-------|--------------|-----------------|------------|---------------|--------------------|-----------------------|----------------|
| Start | 4 | 16 | 1.0 | 2.0 | $1e^{-4}$ | 1.0 | 1.0 |
| End | 4 | 16 | 0.1 | 0.5 | $1e^{-4}$ | 0.1 | 0.1 |

the unsupervised classification ability. We can see that the decoder map generated is partitioned into 16 blocks which correspond to the size of the top layer map. Each block is again having 16 columns corresponding to each top layer column 5.

**Limitation**    As we can see in Fig 8, it's possible that some input handwritten numbers are routed to the wrong columns. This problem could be solved by adding more layers, increasing the size of the map or casting the network into a CNN structure in future research.

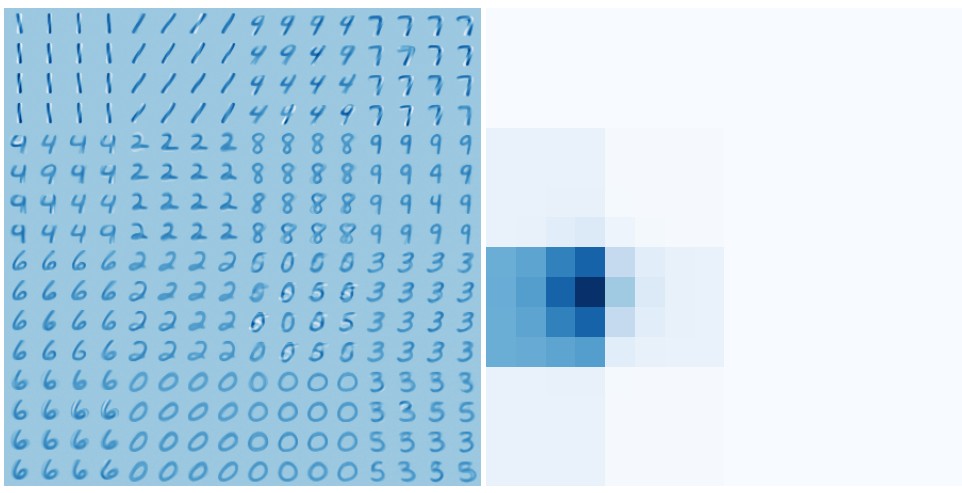

(a) Decoder map with coherence update      (b) Overall neighborhood function

Figure 5: Two layer experiments result. We can see the decoder map is partitioned into $4 \times 4$ blocks.



Figure 6: Selected top encoder map

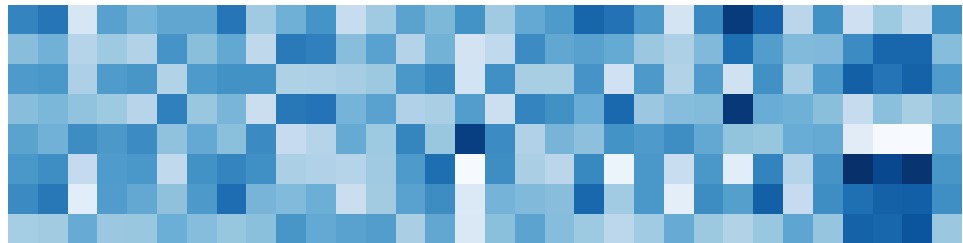

Figure 7: Selected bottom encoder column

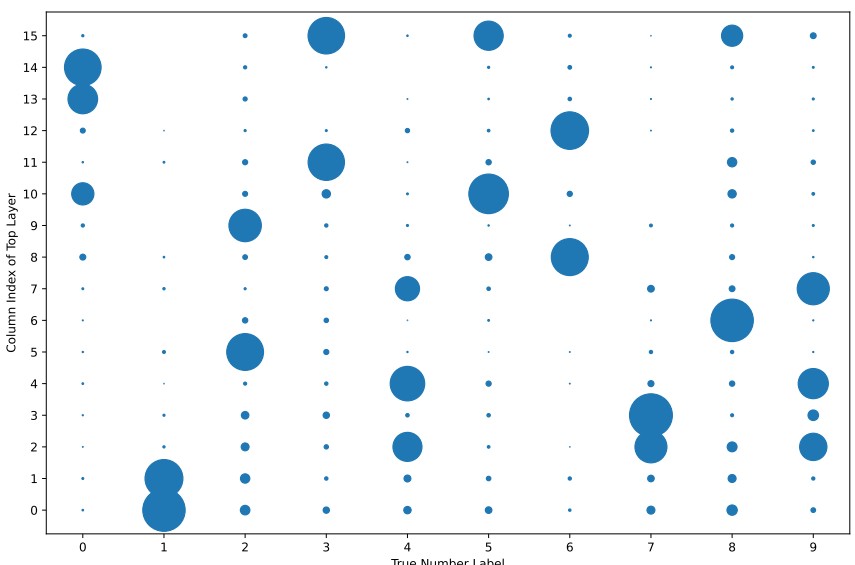

Figure 8: Unsupervised classification. Horizontal axis is true number label, vertical axis is the winning top layer column index. Circle sizes reflect the possibility which the input number is classified to.

## 5 Discussion

**Bus structure and bandwidth saving** The location of the BMU of each layer makes the classification while the vector of the BMU column passed to the next layer attains the details within this class which means that the information not belonging to the current class has been got rid of. Unlike traditional deep neural networks whose feature vector dimension always gets bigger when the network get deeper, in our architecture, the feature vector will always gets smaller with the accompanying BMU classification.

The auxiliary classifier of each layer determines which column has the priority to output to the next layer which resembles a bus structure in the computer system. This point of view gave us an inspiration that the methodology of computer system design could also apply to neural network system design at least to some extend.

**Winner take all competition is too strict, we need a finer competition algorithm**   One limitation of our work is that in the method section all the competition strategies we used are winner take all (WTA) which means only one node is selected as the BMU while all others lose. WTA is a special case of the mutually inhibitory neural algorithm. Normally after mutual inhibition, there will be multiple BMUs[21][22] which means we can build a more complicated structure in the future.

**Neural columns can latch into each other just like the flip-flop circuits**   by mutual inhibition and lateral excitation[23][24]. With neural columns as the functional unit, mutually latched columns by some certain trace can be recalled as a functional group that optimized for some function domain while with another trace we can recall another functional group for another function domain. This means it's possible for us to develop a universal architecture for the human-like neural network machine.

# 6   Conclusion

A novel neural network architecture of functional self organizing map is presented in this paper. By introducing competition and coherence update to a columnar network we can route different columns to each other dynamically. We conducted experiments on a handwritten digits dataset and verified that our method inherited the self organizing ability of the self organizing map, especially on two layered case, we can generate a hierarchically topologically related map which have a unsupervised classification ability. This result suggest that we successfully mapped the functional space of the auto-encoder to a topological space. As for future work, it is interesting to adopt this method to a convolutional neural network or other fine grained model for better regression ability. Finally, we hope this novel architecture provide a new substrate for neural network research which could bring us more advanced artificial intelligence.

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
