# A  Appendix

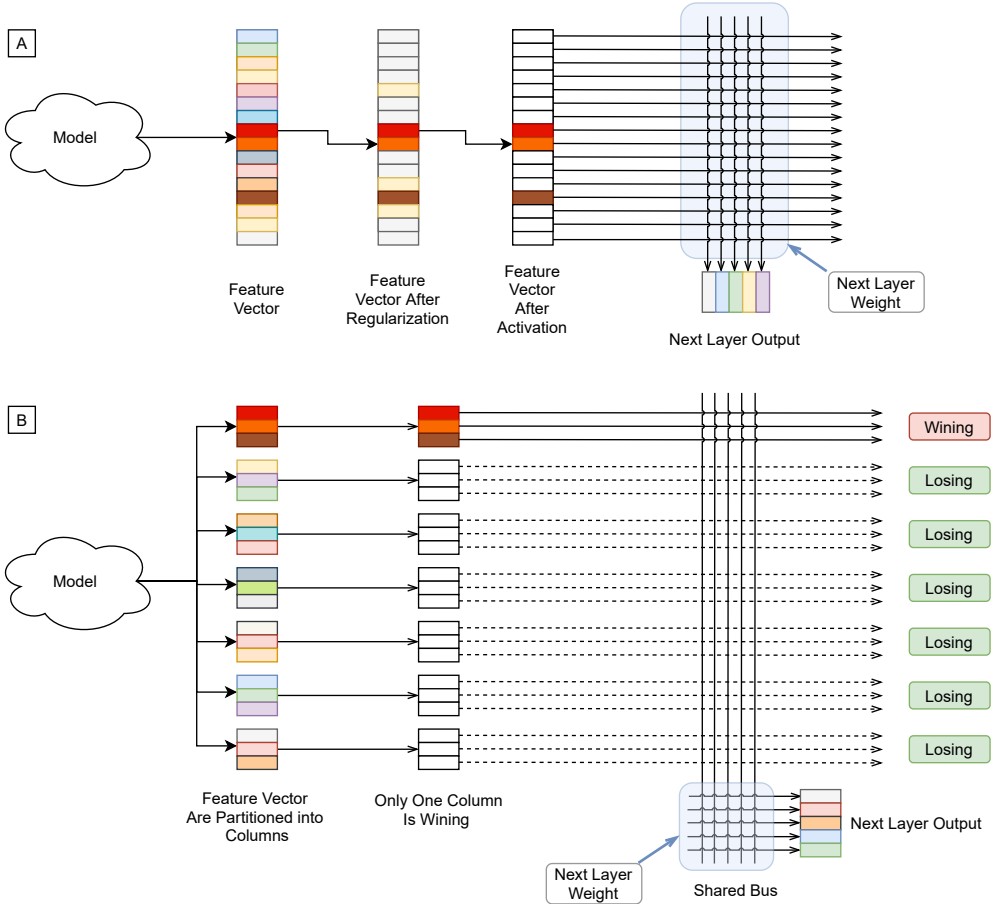

Figure 1: A) After regularization and ReLU activation, most of the output vector of the network model is zero which is indicated by white cells in the feature vector. Although they are zero, they are still needed as the input to the next layer. B) The output of the network model is partitioned into columns, after the competition only one column will win the opportunity to output. Then the size of the next layer is dramatically reduced for its only connected to the shared bus.