# OpenReview forum: "Unsupervised Dynamic Routing Via Competition Over Network Loss"
_NeurIPS.cc/2021/Conference — NeurIPS 2021 Submitted_

### Official Review · Reviewer_H5F1 · 2021-07-16

**Rating:** 4
**Confidence:** 4

**Summary:**

The paper proposed a novel network architecture called functional self-organizing map (fSOM) for making connections between network layers and also a method to discover the connection target automatically by dynamic routing among the neural columns between layers by mutual competition between network columns. The authors presented the results of experiments on a handwritten digits dataset and verified that the method inherited the self-organizing ability of the self-organizing map, especially on the two-layered case, we can generate a hierarchically topologically related map that has an unsupervised classification ability.

**Limitations And Societal Impact:**

Some limitations are discussed. This research doesn't seem to have a societal impact.

**Main Review:**

The paper proposed a novel network architecture called functional self-organizing map (fSOM) for making connections between network layers and also a method to discover the connection target automatically by dynamic routing among the neural columns between layers by mutual competition between network columns. The authors presented the results of experiments on a handwritten digits dataset and verified that the method inherited the self-organizing ability of the self-organizing map, especially on the two-layered case, we can generate a hierarchically topologically related map that has an unsupervised classification ability.

The presented idea is quite original and the description is rather clear. However, the quality of the paper is not sufficient for NeurIPS. Since the experiments were conducted only on a relatively simple (to contemporary machine learning methods) dataset (mnist) and the method was not compared with any other approaches / algorithms, it is hard to assess its significance. Running experiments only on MNIST suggest that it is still relatively early-stage research and further investigation is required. It is possible that for more complex tasks and datasets this method will not be as successful.

Also, there is a link to a Github repository that was created but there is no code the verify or reproduce the results. The last commit was approximately 2 months ago, so it looks that the authors had sufficient time to submit the code used for experiments.

Also, there are many typos / language mistakes, e.g.,:
- "is always fascinating scientists" -> "always fascinates scientists"
- "the "pixels" of the sequence has a correlation of each other" -> "the "pixels" of the sequence have a correlation of each other"
- "the concept of capsules which is a multi-layer columnar network which are trained"-> "the concept of capsules which is a multi-layer columnar network which is trained" (or "the concept of capsules which are multi-layer columnar networks which are trained")
- "for making connection between" -> "for making a connection between"
- "if the feature size is large then the mutual connection matrix" -> "if the feature size is large, then the mutual connection matrix"
- "is the same as in Equation 1 2" -> "is the same as in Equations 1 and 2."
- "map which have a unsupervised classification ability" -> "map which has an unsupervised classification ability"

All in all, the idea seems to be interesting and novel but it requires further investigation and at the current stage, the article is not good enough to be published at NeurIPS. However, it might be appropriate for other conferences.

**Time Spent Reviewing:**

3

---

### Official Review · Reviewer_DBew · 2021-07-19

**Rating:** 3
**Confidence:** 3

**Summary:**

This paper develops a 2D mapping method by extending the self-organizing map (SOM) to a multi-layer version. While the simple version of SOM can be regarded as a single-layer network, the proposed functional SOM (fSOM) adopts the multilayered encoder-decoder architecture. The training technique, called weight coherence update, is introduced to obtain consistent neighborhood structures. The proposed fSOM is applied to the 2D embedding problem using the MNIST dataset.

**Limitations And Societal Impact:**

This paper does not provide a comparison between the proposed method and the existing methods. It is not clear the advantages and disadvantages of the proposed method compared to other baselines. Therefore, the authors should conduct the experimental comparison to reveal the limitations of the proposed method.

**Main Review:**

This paper proposes an extension of the self-organizing map (SOM). The research direction that develops a deep SOM architecture seems to be interesting.

In the equation in section 3.2, the operation in the encoder and decoder seems to be linear projection. There is no activation function, meaning that no non-linear mapping exists in the proposed architecture. Why don't the authors introduce non-linearity into the proposed architecture?

Moreover, the experimental comparison to existing methods is not conducted. The reviewer cannot recognize the advantage of the proposed method against naive SOM and other unsupervised learning methods.

This paper has a lot of unclear points. For example, what is the contribution of an auxiliary classifier? How to determine the hyperparameter in the proposed method? How sensitive is the experimental result against the hyperparameter setting?

There exist several deep SOM methods (see the following literature). The authors should elaborate on the difference between the proposed fSOM and existing methods and the advantages of fSOM.

N. Liu, J. Wang and Y. Gong, "Deep Self-Organizing Map for visual classification," 2015 International Joint Conference on Neural Networks (IJCNN), 2015, pp. 1-6, doi: 10.1109/IJCNN.2015.7280357.

C. S. Wickramasinghe, K. Amarasinghe, D. Marino and M. Manic, "Deep Self-Organizing Maps for Visual Data Mining," 2018 11th International Conference on Human System Interaction (HSI), 2018, pp. 304-310, doi: 10.1109/HSI.2018.8430845.

Vincent Fortuin, Matthias Hüser, Francesco Locatello, Heiko Strathmann, Gunnar Rätsch, "SOM-VAE: Interpretable Discrete Representation Learning on Time Series," ICLR 2019

Florent Forest, Mustapha Lebbah, Hanene Azzag and Jérôme Lacaille (2019). Deep Embedded SOM: Joint Representation Learning and Self-Organization. In European Symposium on Artificial Neural Networks, Computational Intelligence and Machine Learning (ESANN 2019).

---
[Comment after rebuttal] The authors did not provide the responses. So, I keep my score.



**Time Spent Reviewing:**

3

---

### Official Review · Reviewer_6pPz · 2021-07-19

**Rating:** 3
**Confidence:** 5

**Summary:**

The paper introduces an approach to function-level clustering. Function-level clustering means that a conditional computation model is optimized (in this case, parallel copies of a neural network), and that the optimization step is grouped by the respective computation path. This is achieved by using a  "self organizing map" that can compose  to select minimal loss computation paths. In more detail, it:
1. computes the function through all of its existing computation paths
2. selects the lowest-loss path
2. optimizes that path and its neighbors only

**Main Review:**

Pros:
- the paper tackles an interesting idea and a research direction arguably fundamental for interpretable ML and biologically plausible networks
- the idea of updating a neighborhood of computational units is novel and interesting, because it could lead to a functional continuum

Cons:
- the paper does not address the similarities of this work and the rich literature on conditional computation. In particular the "Routing" paradigm (https://arxiv.org/pdf/1904.12774.pdf) is an extremely similar and relevant approach to combining computation paths. "Dispatched Routing Networks" (https://nlp.stanford.edu/projects/sci/dispatcher.pdf) even explicitly models routing and conditional computation as function-level clustering, and discusses several additional aspects not mentioned in the paper
- the paper is poor on experiments; neither does it fully evaluate how the computation paths develop or how they relate, nor does it evaluate on multiple datasets

Originality:
Unfortunately, the authors do not seem to be aware of the existing literature. Given the existing highly relevant (and richer) literature, however, originality of this work is low.

Quality:
The quality of the submission is ok. There are some minor issues around language, but overall the paper is well written, well structured, and has well-thought out illustrations.

Clarity:
The clarity of the submission is ok. There are some minor issues when the authors move from single layer routing to multi-layer routing, and it's not immediately clear how one relates to the other.

Significance:
The area of study - modular neural architectures - is highly relevant for several subfields of ML and the brain sciences. Unfortunately, the limited scope of their experiments (and the low originality) strongly lowers the overall significance.

**Time Spent Reviewing:**

3

---

### Decision · Program_Chairs · 2021-09-27

**Decision:**

Reject

**Comment:**

All the reviewers suggested reject and the authors did not provide any answers to all the questions raised.